# Temperature Dependence of Electrical Resistance in Graphite Films Deposited on Glass and Low-Density Polyethylene by Spray Technology

Angela Longo [1], Antonio Di Bartolomeo [2,3], Enver Faella [2,3], Aniello Pelella [2,3], Filippo Giubileo [3], Andrea Sorrentino [1], Mariano Palomba [1,*], Gianfranco Carotenuto [1], Gianni Barucca [4], Alberto Tagliaferro [5] and Ubaldo Coscia [6,*]

[1] Institute for Polymers, Composites and Biomaterials—National Research Council (IPCB—CNR), SS Napoli/Portici, Piazzale Enrico Fermi, 1-80055 Portici, Italy
[2] Department of Physics "E. R. Caianello", University of Salerno, Via Giovanni Paolo II, 132, 84084 Fisciano, Italy
[3] Superconducting and Other Innovative Materials and Devices Institute—National Research Council (SPIN-CNR), Via Giovanni Paolo II, 132, 84084 Fisciano, Italy
[4] Department SIMAU, Polytechnic University of Marche, Via Brecce Bianche, l-60131 Ancona, Italy
[5] Department of Applied Science and Technology, Politecnico di Torino, Corso Duca degli Abruzzi, 24, 10129 Torino, Italy
[6] Department of Physics 'Ettore Pancini', University of Naples 'Federico II', Via Cintia, l-80126 Napoli, Italy
* Correspondence: mariano.palomba@cnr.it (M.P.); ubaldo.coscia@unina.it (U.C.)

**Abstract:** Graphite lacquer was simply sprayed on glass and low-density polyethylene (LDPE) substrates to obtain large area films. Scanning Electron Microscopy (SEM) images, Raman spectra, X Ray Diffraction (XRD) spectra and current-voltage characteristics show that at room temperature, the as-deposited films on different substrates have similar morphological, structural and electrical properties. The morphological characterization reveals that the films are made of overlapped graphite platelets (GP), each composed of nanoplatelets with average sizes of a few tens of nanometers and about forty graphene layers. The thermoresistive properties of the GP films deposited on the different substrates and investigated in the temperature range from 20 to 120 °C show very different behaviors. For glass substrate, the resistance of the film decreases monotonically as a function of temperature by 7%; for LDPE substrate, the film resistance firstly increases more than one order of magnitude in the 20–100 °C range, then suddenly decreases to a temperature between 105 and 115 °C. These trends are related to the thermal expansion properties of the substrates and, for LDPE, also to the phase transitions occurring in the investigated temperature range, as evidenced by differential scanning calorimetry measurements.

**Keywords:** graphite platelet coatings; low-density polyethylene; thermal expansion coefficient; phase transitions; thermoresistive properties

## 1. Introduction

The improvement of the electrical conductivity of polymers for their use in flexible electronics [1–6] can be achieved efficiently by coupling them with materials such as carbon black [7], carbon fibers [8–10], graphite [11,12], carbon nanotubes [13], graphene [14,15] or reduced graphene oxide [16,17]. In particular, it is possible to fabricate electrically conductive paths [18], printed radio frequency devices [19], flexible sensors [2,20] and piezoresistive sensors [21] by covering the surfaces of polyethylene terephthalate, low-density polyethylene (LDPE) or poly (methyl methacrylate) with graphite or graphene layers [22–24].

These carbon-based layers can be obtained by using different techniques such as the casting and drying of inks [18], chemical vapor deposition [25], micromechanical techniques

utilizing the spreading of graphite nanoplatelets in an alcoholic suspension [22–24] or spraying conductive composites [26,27]. Above all, this last technique is simple, low cost, and suitable for the fabrication of large area conductive films [28] and polymeric film strain gauges suitable for the measurement of large elongations [26].

In this paper, an extensive study is conducted on the properties of graphite platelet (GP) films obtained by spraying a commercial lacquer on different substrates. The structural and morphological characterizations of GP films are performed by a variety of techniques such as differential scanning calorimetry (DSC), X-ray diffraction (XRD), scanning electron microscopy (SEM) and Raman spectroscopy.

The thermoresistive properties of GP deposited on LDPE and glass substrates are investigated by the resistance trends in the 20–120 °C temperature range. In the case of films on LDPE, variations in the thermal expansion coefficient and phase transitions occurring in the polymer substrates are taken into account in the discussion. Understanding the dependence of thermoresistive properties on the thermomechanical ones is crucial in view of large-scale production.

## 2. Materials and Methods

Using Graphit 33, a commercial lacquer produced by kontakt Chemie (Zelecity, Belgium), spray technology was exploited to deposit large area graphite-based films on glass and LDPE substrates. LDPE substrates of size 15 cm $\times$ 15 cm $\times$ 90 µm were produced by Sabic (Riyadh, Saudi Arabia, 99.77% by weight, Mn = 280,000 g·mol$^{-1}$, melt flow index of 7 g/10 min and crystalline fraction (Xc) of 32.8%), while glass substrates (SiO$_2$) of size 2.5 cm $\times$ 7.6 cm $\times$ 1 mm were fabricated by Pearl, Inc. (West Hollywood, CA, USA). Figure 1 shows the scheme of the apparatus used for the spray technique.

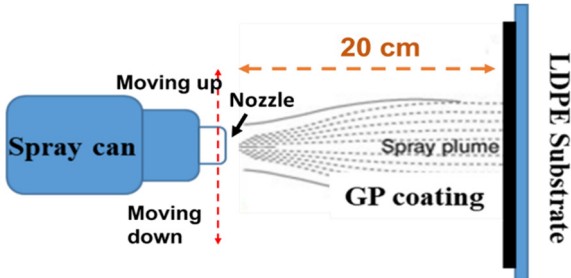

**Figure 1.** Scheme of the apparatus used for the spray deposition technique.

The spray nozzle was directed to a distance of 20 cm from the plane containing the substrates to produce a uniform deposition by full cone spot and horizontally moved so that a graphite film could completely cover the surface of the substrates. Four runs of depositions were performed, and for each run, a pair of graphite films was obtained at room temperature (RT) under the same deposition conditions on glass and LDPE substrates.

Before lacquer spraying, no cleaning treatment was performed on the LDPE substrates, while glass substrates were cleaned by acetone. After spraying, the coated substrates were kept in air at room temperature to dry for 4 h.

The graphite content in the Graphite 33 lacquer measured by TGA analysis corresponded to 82% by weight, as indicated in our previous article (see Figure 2 in Ref. [27]) and the as deposited amount of lacquer equal to 7.20 mg/cm$^2$ became 0.983 mg/cm$^2$ (0.806 mg/cm$^2$ of graphite) after drying in air for 4 h.

A Raman spectrometer, InviaH-Renishaw (New Mils, Wotton-under Edge, Glowcester Shire, GL128JR, UK), was employed to acquire the Raman spectra of the GP films on both glass and LDPE substrates. The analysis was carried out using a green argon laser with 514.5 nm wavelength and about 2 µm diameter beam spot. The microscope was operated at 50$\times$ magnification and 10 s exposure time. Using $\sim$5 mW laser power (corresponding to 5% of the available laser power), 100 to 3500 cm$^{-1}$ scans were performed. The quantum efficiency of the detector was corrected by a dedicated MATLAB software

(Matlab 9.7.0. 1190202 R2019b, Mathworks Inc., Natick, MA, USA), which was also used for the baseline subtraction and the data processing.

The analysis of the samples surface was performed by scanning electron microscopy by means of a Zeiss Supra 40 field-emission SEM (Carl Zeiss Microscopy GmbH, Jena, Germany). The samples were accurately attached on stubs and scrutinized without any preparation. As graphite-based materials are conductive, their surface was electrically grounded.

The mean thickness value of the investigated films, estimated by transmission electron microscopy observations, as reported elsewhere [23], was about 2.5 μm.

Sample structural properties were also studied by X-ray diffraction (XRD) measurements performed by a Panalytical, X'PERT PRO diffractometer (Malvern Panalytical, Cambridge, UK) with λ = 0.154 nm CuKα radiation, 5°–60° 2θ range, 0.0130° scan step and 18.9 s full scan time.

Phase transitions of the LDPE substrate were investigated by differential scanning calorimetry (DSC) using a Q1000-TA instrument (TA instrument, New Castle, PA, USA). The apparatus was calibrated using indium and zinc as reference and non-isothermal cooling-heating-cooling tests were carried out at different scan rates in the 20–180 °C temperature range.

A fully automated GNOMIX high-pressure mercury dilatometer was used for Pressure-Volume-Temperature (PVT) experiments to explore the thermal expansion properties of the LDPE substrate [29]. The instrument was able to detect changes in specific volume with an absolute accuracy of 0.004 $cm^3 \cdot g^{-1}$. The polymer density variations were measured from room temperature to 137 °C with the scanning rate of 3 °C/min. The thermal linear strain, $\varepsilon$, of the LDPE was determined in heating mode from the changes in volume obtained during isobaric PVT runs as:

$$\varepsilon = \upsilon \times (V - V_0)/V_0 \tag{1}$$

where $\upsilon$ is the Poisson modulus equal to 0.46 for LDPE, $V_0$ is the initial volume of the sample at the temperature $T_0 = 20$ °C and V is the volume of the sample at the temperature, T. Resistance measurements of the deposited graphite film versus temperature, in the 20–120 °C range, were performed under 2 mbar vacuum in a coplanar configuration, after spreading the sample surfaces by 1 cm long and 1 mm spaced silver-paint contacts.

A source-measurement unit (SMU) Keithley 4200-SCS (Tektronix, Inc., Beaverton, OR, USA), electrically connected to the four micromanipulators of a Janis Research ST-500 probe station (Janis Research, Woburn, MA, USA), was used to measure the resistances of the samples as mean values from the I-V characteristics taken at fixed temperatures.

## 3. Results

Films deposited on different substrates by spraying Graphit 33 lacquer under the same preparation conditions were investigated for their morphological, structural and thermoresistive properties. After spraying the lacquer, a good adhesion of the graphite film takes place on LDPE substrate because graphite platelets interact with the surface of non-polar materials, such as polyethylene. On the other hand, the polymeric binder present in the lacquer allows the adhesion of the graphite film to the glass substrate.

The Raman spectra of two representative samples prepared on glass and LDPE substrates are plotted in Figure 2a,b, respectively. No substrate signals are detected as the films' thicknesses are higher than the penetration length of the laser radiation used in the Raman apparatus, i.e., a few hundred nanometers. Hence, the Raman spectra are due to the graphite platelets embedded in the lacquer. The features in the 1000–1800 $cm^{-1}$ region are quite similar in the two spectra: a narrow D peak, a sharp and narrow G band with a shoulder (D*) due to the presence of defects [30]. The peaks are sharp, indicating that the graphite platelets are of good crystalline quality. The peak intensity ratio, $I_d/I_g$, averaged over the sampled points, are 0.55 and 0.80 for the films on glass and LDPE, respectively. It is worthwhile to note that the D peak is generated by the platelet edges [31], and indeed in the case of a platelet slightly larger than the beam and perpendicular to it, no edges will be excited and no D peak will be detected [31]. On the other end, for bended platelet

not perpendicular to the beam, some edges will be illuminated producing a D peak. The results indicate that more edges fall into the area swept by the laser beam in the LDPE case. This is consistent with a slightly more disorder in orientation of the platelets on the LDPE substrate.

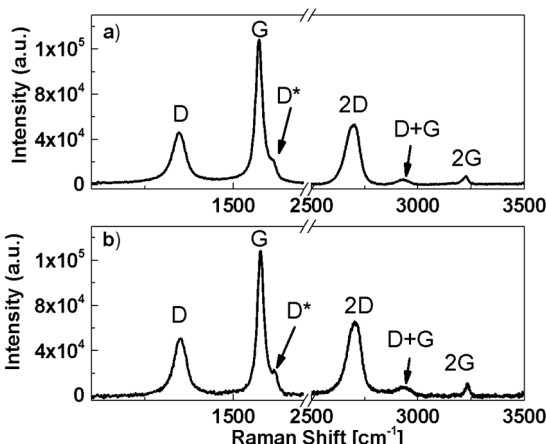

**Figure 2.** Raman spectra of films deposited by Graphit 33 lacquer on glass (**a**) and LDPE (**b**).

Three further peaks can be detected at higher wavenumbers between 2500 and 3500 cm$^{-1}$ in both spectra. The intensity of the left shouldered 2D peak indicates a multilayered structure, while the D + G peak and the 2G peak have the typical width and intensity of graphite material.

The surface morphology of the samples was investigated by SEM. The micrographs of two films deposited on glass and LDPE substrates are shown in Figure 3a,b, respectively. SEM micrographs reveal that the films on both substrates are composed of aggregated graphite platelets (GP) having a tendency to align parallel to the LDPE/film interface but arranged in a rather disordered way and partially overlapping. GP are irregular in shape, with a broad distribution of lateral dimensions ranging from a few hundreds of nanometers to a few micrometers. Although the platelets tend to align parallel to the interfacial plane, it can be observed that platelets tilted with respect to the substrate plane (as in the insets of Figure 3) make the surface of the films rough and porous. SEM analysis reveals that films deposited on the different substrates exhibit a similar surface structure.

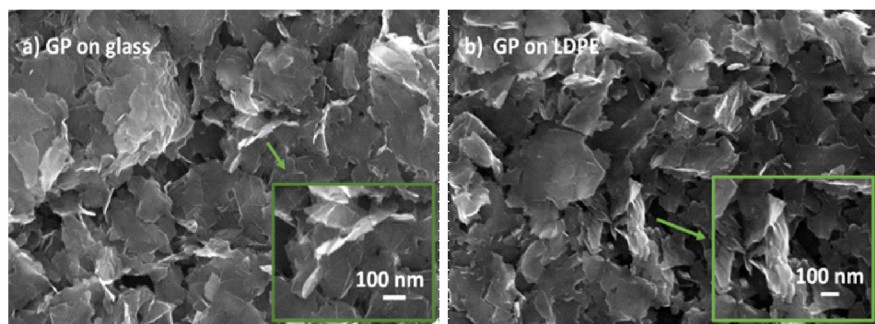

**Figure 3.** SEM micrograph of GP films deposited on glass (**a**) and LDPE (**b**). The insets show a magnification of platelets tilted with respect to the substrate planar surface.

Further information on the structure of the GP films is obtained through XRD measurements. The XRD spectrum of the film deposited on glass is shown in Figure 4a. It includes only one peak centered at 2θ = 26.46, attributed to the (002) crystallographic planes of graphite and the typical amorphous halo due to the glass substrate [International Center for diffraction Data file: 00-008-0415].

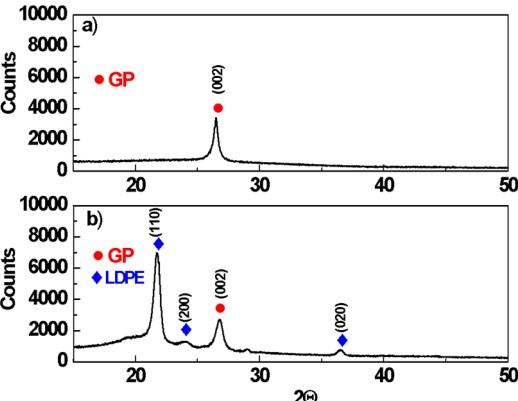

**Figure 4.** XRD diffractograms of graphite platelet (GP) films deposited on glass (**a**) and LDPE (**b**).

On the other hand, the XRD diffractogram of the film deposited on LDPE and displayed in Figure 4b includes both LDPE and graphite diffraction peaks. Three peaks assigned to the crystalline phase of LDPE [32], superimposed on a large halo generated by the amorphous polyethylene phase [33,34], can be identified: a dominant sharp peak at $2\theta$ = 21.79°, a weak broad peak at $2\theta$ = 24.05° and one more weak peak at $2\theta$ = 36.57° corresponding, respectively, to (110), (200) and (020) crystallographic planes. In addition, a peak of low intensity at $2\theta$ = 26.74° due to the graphite (002) plane [ICDD: 75-1621] is also detected.

The average thickness of the graphite crystals, $L_{(hkl)}$, perpendicular to the lattice plane identified by Miller's indices (hkl), is evaluated using Scherrer's equation:

$$L_{(hkl)} = K \times \lambda / (FWHM \times \cos(\theta)) \quad (2)$$

where $\theta$ is the half of the corresponding scattering angle of peak (hkl), FWHM in radians is the full-width at half-maximum and stems from the Gaussian fit of the (hkl) peak, K assumed to be 0.89 is a constant dependent on the crystallite shape [35,36], and $\lambda$ is the wavelength of the X-ray radiation (Cu-K$_{\alpha 1}$ = 0.15481 nm). The Equation (2) applied to the (002) peak of graphite yields $L_{(hkl)}$ = 15.2 ± 0.5 nm and $L_{(hkl)}$ = 12.6 ± 0.4 nm for films deposited on glass and LDPE, respectively. Moreover, the inter-plane distance is obtained by Bragg's law for a family of lattice planes (hkl) [37,38]:

$$d_{(hkl)} = n\lambda / \sin(\theta) \quad (3)$$

where n = 1 is the diffraction peak order. The calculated values are $d_{(002)}$ = 0.337 ± 0.002 nm and $d_{(002)}$ = 0.333 ± 0.003 nm for the films on glass and LDPE, respectively, in agreement with the separation distance between two graphene layers (d = 0.335 nm) inside the graphite phase [39].

Finally, the average number of the graphene layers (N) per crystalline domain is determined from the following equation:

$$N = (L_{(002)} / d_{(002)}) + 1 \quad (4)$$

The calculated N values are 46 ± 1 and 38 ± 1 for the samples deposited on glass and LDPE, respectively. Thus, it can be stated that the deposited films are made of an aggregate of overlapping graphite platelets, each formed by nanoplatelets with average sizes of a few tens of nanometers and about forty graphene layers.

Regarding the thermoresistive properties of GP films, the influence of the substrate thermomechanical parameters has to be considered. Since the properties of glass are well known, only the temperature ranges, in which phase transitions of LDPE occur and the coefficient of linear thermal expansion are determined in this work.

Figure 5 shows the conventional DSC results from the heating of a representative LDPE substrate in the 20–180 °C range at the scanning rate of 10 °C/min.

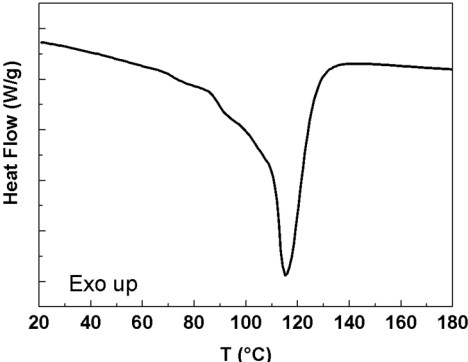

**Figure 5.** DSC heating curves of a LDPE sample in the 20–180 °C temperature, T, range at the rate of 10 °C/min.

LDPE is a ductile and flexible material at room temperature. It is solid and stable in the temperature range from 20 to 85 °C, while it presents a complex phenomenon resulting from a broad melting transition and reorganization of the crystalline phase if the temperature is greater than 85 °C and less than or equal to 120 °C, as displayed in Figure 4 by the enthalpic curve having a minimum value at about 114 °C. At temperatures above 120 °C, the LDPE first becomes "sticky" (120–140 °C) and then a liquid melt (140–180 °C). As with any other polymeric material, this temperature range is characterized by a partial rearrangement of the molecular chains with the formation of a high viscous liquid with some elastic characteristics (viscoelastic behavior). During this phase transition, the material becomes sticky because it has some typical characteristics of liquids and therefore "wets" the surfaces with which it is in contact; but then, due to its elasticity, it will resist separation when stressed [40].

The thermal expansion properties of this polymer are studied by means of the thermal linear strain, ε, of a typical LDPE substrate plotted, in Figure 6, as a function of the temperature, T, in the 20–137 °C range. Here, ε is defined as:

$$\varepsilon = (L - L_0)/L_0 \tag{5}$$

where $L_0$ is the initial length of the sample at the temperature $T_0$ = 20 °C and L is the length of the sample at the temperature, T.

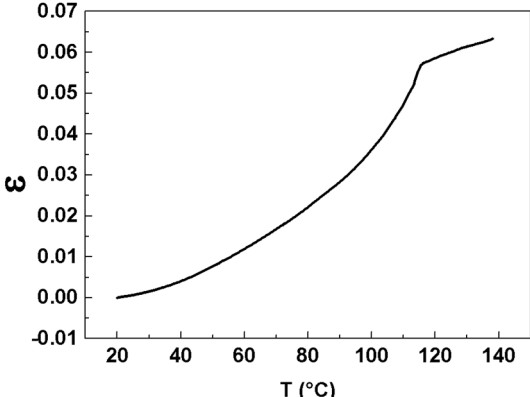

**Figure 6.** Linear strain ε vs temperature, T, of an LDPE substrate.

Clearly, the LDPE thermal expansion is not proportional to the change in temperature, and therefore the linear coefficient of the thermal expansion, CTE, at a given temperature T is calculated as:

$$CTE = \frac{1}{L}\frac{dL}{dT} \qquad (6)$$

where L is the length of the sample at the temperature T and dL/dT is the rate of change of the linear dimension per unit change in temperature.

The CTE versus T plot is reported in Figure 7.

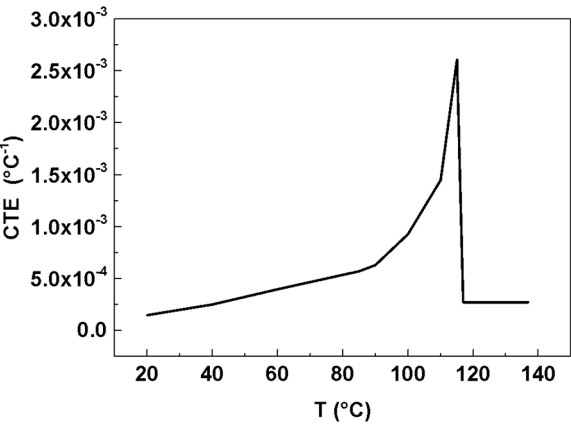

**Figure 7.** The coefficient of linear thermal expansion, CTE, vs. temperature, T, of an LDPE substrate.

The CTE at $T_0$ = 20 °C is $1.5 \times 10^{-4}$ °C$^{-1}$ in agreement with data in the literature [41]. Furthermore, by correlating the CTE trend with the DSC results one obtains that in the 20–85 °C range, where the LDPE phase is solid and stable, the CTE increases between $1.5 \times 10^{-4}$ and $5.7 \times 10^{-4}$ °C$^{-1}$, while if the temperature is greater than 85 °C, where a broad melting transition takes place, CTE increases faster, up to $2.6 \times 10^{-3}$ °C$^{-1}$ near 114 °C$^{-1}$. Finally, the CTE suddenly drops to a constant value of $2.7 \times 10^{-4}$ °C$^{-1}$ from 117 to 137 °C.

In the investigated temperature range, this polymer is an excellent insulator with good dielectric properties and a high-volume resistance, but its surface can be easily made electrically conductive by depositing a GP film by spraying Graphit 33 lacquer, as shown in Figure 8, which displays the current-voltage (I-V) characteristics of a LDPE sample sprayed with Graphit 33 lacquer. The measurements are performed in vacuum in two-probe configuration at the temperature of 20 °C. Figure 8 also shows the I-V characteristics of a GP film produced under the same deposition conditions on glass substrate.

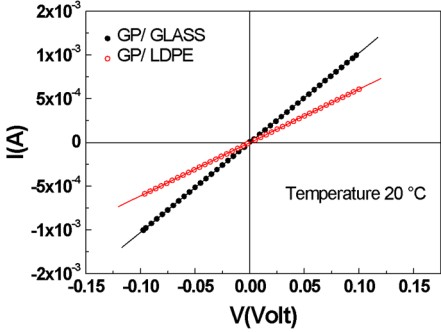

**Figure 8.** I-V characteristics of the GP films deposited on glass (black circle symbols) and LDPE (red circle symbols) substrates. The straight lines fit the experimental data.

The I-V characteristics of both samples are linear, indicating ohmic contacts. The resistance values, $R_0$, resulting from the best fit of the measured I-V curve with a correlation coefficient r = 1, are (97.49 ± 0.02) Ω for GP deposited on glass and (164.00 ± 0.03) Ω for GP on LDPE.

The thermoresistive properties of the GP samples are explored by measuring the resistance R while raising the temperature from 20 to 120 °C. In Figure 9, the $R/R_0$ ratios are displayed as a function of the temperature, T.

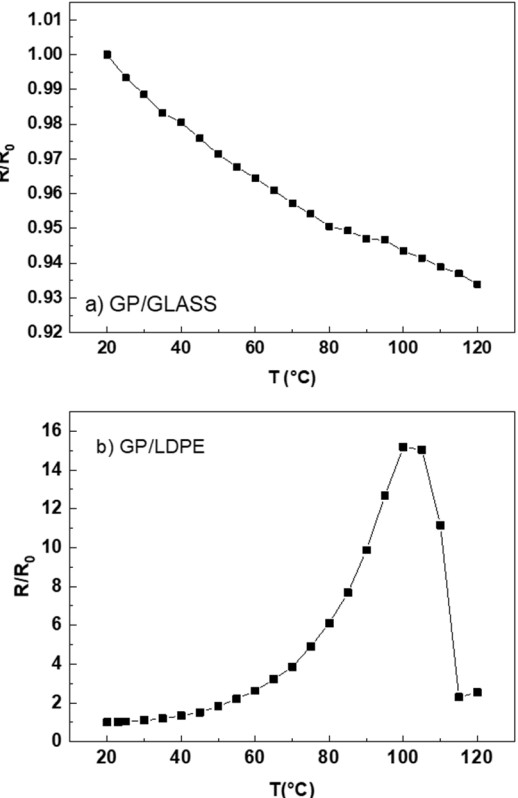

**Figure 9.** Ratio of the resistance R to the initial resistance $R_0$ at 20 °C ($R/R_0$) vs. temperature, T, for GP films deposited on glass (**a**) and LDPE (**b**) substrates. Black lines are guides.

Different thermoresistive behaviors can be noted. An increasing temperature corresponds to a decreasing resistance of GP film on glass (Figure 9a). Conversely, for GP on LDPE, the resistance rises strongly, more than one order of magnitude, in the range from 20 °C to about 100 °C, and after reaching the maximum value, it suddenly decreases between 105 and 115 °C (Figure 9b). These trends can be ascribed to the different thermomechanical properties of the substrates. Indeed, glass has a linear thermal expansion coefficient, CTE = $6 - 9 \times 10^{-6}$ °C$^{-1}$ [42] close to that of graphite ($4 - 8 \times 10^{-6}$ °C$^{-1}$ [42]); therefore, the thermal expansion properties of the substrate do not affect those of the film, the resistance of which decreases with the rising to T, as in graphite [43].

On the other hand, as determined by the thermal expansion experiments, the CTE of the LDPE substrate at $T_0 = 20$ °C is $1.5 \times 10^{-6}$ °C$^{-1}$, thus more than one order of magnitude greater than in graphite, and it increases further as a function of temperature up to 114 °C, and then CTE assumes a constant value of $2.7 \times 10^{-6}$ °C$^{-1}$ between 117 and 137 °C.

Interestingly, from 20 to 100 °C, the trend of resistance ratio is correlated with the increasing mismatch between LDPE substrate and GP film because in the 20–85 °C temperature range $R/R_0$ increases slowly as the CTE of LDPE, while if the temperature is greater than 85 °C and less than or equal to 100 °C, the slope of the $R/R_0$ curve increases sharply as in the case of CTE-T graph shown in Figure 7.

Consequently, the resistance increase observed in the 20–100 °C range, as visible in Figure 9b, can be attributed to the thermal expansion coefficient mismatch between the LDPE and GP films.

Indeed, the higher thermal expansion of the polymeric substrate can induce strains in the deposited film that decrease the contact area among the platelets and cause the

occurrence of micro/nanofractures. These occurrences reduce the number of conductive paths and the film resistance increases.

The decrease in resistance between 105 and 115° C, despite the increase in the polymer substrate's thermal expansion, can be due to the fact that in this range a significant fraction of the substrate has melted, resulting in a sharp decrease in the mechanical modulus of the LDPE which favors the sliding of the graphite platelets and their redistribution with the formation of new conductive paths on the sample surfaces and the consequent decrease in the film resistance.

The electrical conductivity of the LDPE filled with carbon fiber composites carbon or black showed a similar temperature dependence [44]. The investigated GP films on LDPE exhibit significant thermoresistive response and have potential applications for flexible electronics, including temperature sensors and self-switching components.

## 4. Conclusions

Large area graphite platelets films were deposited on glass and LDPE substrates by a spray technology using a commercial graphite lacquer. Analyses of Raman spectra, SEM images and XRD diffractograms show that the films are made of overlapped graphite platelets each composed of nanoplatelets with average size of a few tens of nanometers and forty graphene layers. The enthalpic curve obtained by DSC measurements of LDPE indicates that this polymer is solid and stable in the 20–85 °C range, while it undergoes a broad melting transition in the temperature range between 85 and 120 °C. In particular, the endothermic peak shows a minimum at about 114 °C. It has been found that the thermal expansion properties of LDPE are strongly dependent on the temperature. Indeed, in the 20–85 ° C range, the CTE increases between $1.5 \times 10^{-4}$ and $5.7 \times 10^{-4}$ °C$^{-1}$, while if the temperature is greater than 85 °C, CTE increases faster, up to $2.6 \times 10^{-3}$ °C$^{-1}$ near 114 °C, and then it drops to a constant value of $2.7 \times 10^{-4}$ °C$^{-1}$ from 117 to 137 °C.

The thermoresistive properties of the GP film are affected by the mismatch of CTE between the film and the substrate. Indeed, the CTE of the glass substrate is close to that of the graphite film, and the thermal expansion properties of the substrate do not influence those of the film, the resistance of which decreases with the increasing of the temperature, as in graphite. Conversely, for GP deposited on LDPE substrate, the increasing CTE mismatch between the film and the substrate, in the 20–100 °C range, induces strains in the film that decreases the contact area among the platelets and increases the nano/micro-fractures, thus resulting in the resistance increase of the films. Finally, between 105 and 115 °C, the melting of the polymer substrate favors the sliding of the graphite platelets with the formation of new conductive paths that decrease the film resistance.

**Author Contributions:** Conceptualization, U.C., G.C., A.S., G.B. and A.D.B.; software, A.L., A.S., G.B. and M.P.; validation, U.C., A.L., M.P., A.S., G.B. and A.T.; formal analysis, A.L., M.P., U.C., A.S. and G.B.; investigation, G.B., E.F., A.P., F.G., A.L., A.S. and M.P.; data curation, G.C., A.L. and M.P.; writing—original draft preparation, G.C., M.P., A.L., A.D.B., G.B., A.S., E.F., A.P., F.G., A.T. and U.C.; writing—review and editing A.L., A.D.B., E.F., A.P., F.G., A.S., M.P., G.C., G.B., A.T. and U.C.; supervision, U.C., A.S., G.C. and A.D.B. All authors have read and agreed to the published version of the manuscript.

**Funding:** This research received no external funding.

**Institutional Review Board Statement:** Not applicable.

**Informed Consent Statement:** Not applicable.

**Data Availability Statement:** Not applicable.

**Acknowledgments:** We wish to acknowledge the contribution of the late Massimo Rovere to this paper, unfortunately one of his last contributions to science.

**Conflicts of Interest:** The authors declare no conflict of interest.

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
