# Peer review of "Temperature Dependence of Electrical Resistance in Graphite Films Deposited on Glass and Low-Density Polyethylene by Spray Technology"

_coatings, doi:10.3390/coatings12101446_

Round 1

Reviewer 1 Report

  1. The work deals with the coating of LDPE and glass surfaces with graphite platelets by lacquer spraying. The samples are characterized by SEM, DRX, Raman and resistance. Additional characterization is given for LDPE support. The relevance of the work is low, the novelty and impact expected should be stated. The procedures are not well described, and characterization has some gaps. I do not recommend publication.
  2. Materials and methods. More information should be given about the graphite concentration in the commercial lacquer and the amount of graphite deposited per sq cm of film.
  3. More precise information on the number of samples and how they were prepared should be given. E.g. in Figure 2 two “representative samples” are analyzed and compared, but there is no information about how they were prepared and whether differences between them can be expected.
  4. Page 3. Deposited films are said to have a graphite-like structure. It is hard to think that the film itself has a graphite-like structure. Figure 2 does not support this comment. Is there a difference between the spectra for graphite sprayed on film and initial graphite in the lacque? The authors may need to revise the whole discussion on Raman results, attributing the signals to the graphite deposited, and not as a specific property of the film.
  5. The same applies to DRX. Figure 2 does not support the comment on a film structure with “overlapping graphete platelets”, arrangement seems to be rather random. Analysis of lacquer may be needed for comparison and assessing if properties and Raman and DRX signal change due to arrangement during sprying and drying.
  6. DSC for LDPE is not relevant, the properties of LDPE are well-known. Interest would come from potential changes in behaviour due to interaction with deposited graphite. The same applies to thermal expansion, etc. in Figures 5 and 6, which seems to be tested only for the LDPE support.

Author Response

Comments on coatings-1844614 manuscript        

Response to Reviewer 1 

1. General comments of Reviewer 1

The work deals with the coating of LDPE and glass surfaces with graphite platelets by lacquer spraying. The samples are characterized by SEM, DRX, Raman and resistance. Additional characterization is given for LDPE support. The relevance of the work is low, the novelty and impact expected should be stated. The procedures are not well described, and characterization has some gaps. I do not recommend publication.

Response to the general comments: We would like to thank the reviewer for the comments. We have improved our manuscript following his/her suggestions. All the modified paragraphs of the text are highlighted in yellow below and in an additional copy of the revised manuscript uploaded as source file coatings-1844614rev.docx

2. Reviewer’s  comment

Materials and methods. More information should be given about the graphite concentration in the commercial lacquer and the amount of graphite deposited per sq cm of film.

Reply

According to the  Reviewer’s  comment more information on the graphite concentration are given

below in the revised text and highlighted in yellow.

Pag. 2 line 25

Previous text

After spraying, the coated substrates were dried in air for 4 h at room temperature.

Revised text

After spraying, the coated substrates were dried in air for 4 h at room temperature.

The graphite content in the Graphite 33 lacquer measured by TGA analysis corresponded to 82% by weight as indicated in our previous article (see Figure 2 in Ref [27]) and the as deposited amount of lacquer  equal to 7.20 mg/cm2, became 0.983 mg/cm2 (0.806 mg/cm2 of graphite) after drying in air for 4h.

3. Reviewer’s  comment

More precise information on the number of samples and how they were prepared should be given. E.g. in Figure 2 two “representative samples” are analyzed and compared, but there is no information about how they were prepared and whether differences between them can be expected.

Reply: We  would like to thank the reviewer for his/her comments that helped us to improve the manuscript. Changes to the text are reported below in the revised text and highlighted in yellow.  As for the differences between the properties of the films deposited on the different substrates, we did not expect such marked differences in the resistance of GP films as a function of temperature as we have stressed in the results and discussion section.

Pag.2 line 22 

Previous text

Large area thin films, about 15 cm x 15 cm of graphite-based material were deposited on glass and LDPE substrates by spray technology, using a commercial lacquer, Graphit 33 (from Kontakt Chemie, Zelecity, Belgium).

Revised text                                              

Large area thin films of graphite-based material were deposited on glass and LDPE polymer substrates by spray technology, using a commercial lacquer, Graphit 33 (from Kontakt Chemie, Zelecity, Belgium).  LDPE substrates, of size 15 cm x 15 cm x 90 mm, were produced by Sabic (Saudi Arabia, 99.77% by weight, Mn=28 0000 gmol−1, melt flow index of 7 g/10 min and crystalline fraction (Xc) of 32.8%),  while glass substrates (SiO2)  of size 2.5 cm x 7.6 cm x 1 mm, were fabricated by Pearl inc. (West Hollywood, CA, USA).

The figure 1 shows the scheme of the apparatus used for the spray technique.

Figure 1. Scheme of the apparatus used for the spray deposition technique.

The spray nozzle was directed to a distance of 20 cm from plane containing the substrates to produce a uniform deposition by full cone spot, and horizontally moved so that a graphite film could completely cover the surface of the substrates. Four runs of depositions were performed and for each run a pair of graphite films was obtained at room temperature (RT) under the same deposition conditions on glass and LDPE substrates.

Before lacquer spraying no cleaning treatment was performed on LDPE substrates, while  glass substrates were cleaning by acetone.

4. Reviewer’s  comment

Page 3. Deposited films are said to have a graphite-like structure. It is hard to think that the film itself has a graphite-like structure. Figure 2 does not support this comment. Is there a difference between the spectra for graphite sprayed on film and initial graphite in the lacque? The authors may need to revise the whole discussion on Raman results, attributing the signals to the graphite deposited, and not as a specific property of the film….. The same applies to DRX.                             

Reply

We thank the referee for pointing out the improper description. No significant difference is observed between the lacquer’s  Raman spectrum (not reported) and the deposited films. We have revised the manuscript with  a new text regarding Raman results to make the discussion clearer. Changes to the text are reported below in the revised text and highlighted in yellow.                 

As regards the XRD measurements of  the films on LDPE substrates, the diffractogram show that the peaks  detected at 2θ = 26.76° are due to the deposited films and can be attributed to the graphite (002) planes [ICDD: 75-1621]. The absence of other less intense diffraction peaks assigned to the (100), (101), (102) and (004) family of lattice planes of the graphite [Klug H P and Alexander L E 1974 “X-Ray Diffraction Procedures for Polycrystalline and Amorphous Materials” 2nd edn (New York: Wiley)], indicates the  iso-orientation tendency (planar arrangement) of graphite platelets in the films.

Pag 3 Line 20

Previous text

The Raman spectra of two representative samples prepared on glass and LDPE substrates are plotted in Figures 1a) and 1b), respectively. 

Spectra analysis denotes that the deposited films have a graphite-like structure with similar features on both substrates: a narrow D peak, a sharp and narrow G band, with a shoulder (D*) due to the presence of defects [30, 31]. The peak intensity ratio, Id/Ig, averaged over the sampled points, are 0.55 and 0.80 for the films on glass and LDPE, respectively.  It reveals that the films are obtained from good quality graphite-based material and exhibit slightly higher defect density when deposited on LDPE.

Three other peaks can be detected in both spectra at higher wavenumbers between 2500 and 3500 cm−1: an intense left shouldered 2D peak indicating a multilayered structure, a D + G peak, and a 2G peak typical in width and intensity of graphite material. No substrate signal is detected indicating that the film’s thickness is higher than the penetration length of the laser radiation used in the Raman apparatus.

Revised text

The Raman spectra of two representative samples prepared on glass and LDPE substrates are plotted in Figures 1a) and 1b), respectively. No substrate signals are detected as the films thicknesses are higher than the penetration length of the laser radiation used in the Raman apparatus, a few hundred nanometers. Hence, the Raman spectra are due to the graphite platelets embedded in the lacquer. The features in the 1000-1800 cm-1 region are quite similar in the two spectra: a narrow D peak, a sharp and narrow G band, with a shoulder (D*) due to the presence of defects [30]. The peaks are sharp, indicating that the graphite platelets are of good crystalline quality. The peak intensity ratio, Id/Ig, averaged over the sampled points, are 0.55 and 0.80 for the films on glass and LDPE, respectively. It is worthwhile to note that the D peak is generated by the platelet edges [31], indeed in the case of a platelet slightly larger than the beam and perpendicular to it, no edges will be excited and no D peak will be detected [31]. On the other end, for bended platelet not perpendicular to the beam, some edges will be illuminated producing a D peak. The results indicate that more edges fall into the area swept by the laser beam in the LDPE case. This is consistent with a slightly more disorder in orientation of the platelets on the LDPE substrate. Three other peaks can be detected in both spectra at higher wavenumbers between 2500 and 3500 cm−1: an intense left shouldered 2D peak indicating a multilayered structure, a D + G peak, and a 2G peak typical in width and intensity of graphite material.

5. Reviewer’s  comment

Figure 2 does not support the comment on a film structure with “overlapping graphete platelets”, arrangement seems to be rather random. Analysis of lacquer may be needed for comparison and assessing if properties and Raman and DRX signal change due to arrangement during sprying and drying.

Reply

We would thank the reviewer for pointing out our improper description on the graphite platelet film morphology.

In order to provide a more appropriate description, the changes to the text are reported below in the revised text and highlighted in yellow. As for the suggestion to compare Raman and DRX signals during spraying and drying of the films, in order to test the arrangement of the GP, we are planning to carry out further investigations in a future work.

Pag.4 line 5

Previous text

SEM micrographs reveal that the films on both substrates are composed of an aggregate of overlapping graphite platelets (GP) irregular in shape, with a broad distribution of lateral dimensions ranging from a few hundred of nanometers to a few micrometers.

Revised text

SEM micrographs reveal that the films on both substrates are composed of aggregated graphite platelets (GP) having a tendency to align parallel to the LDPE/film interface but arranged in a rather disordered way and partially overlapping. GP are irregular in shape, with a broad distribution of lateral dimensions ranging from a few hundred of nanometers to a few micrometers.

6. Reviewer’s  comment

DSC for LDPE is not relevant, the properties of LDPE are well-known. Interest would come from potential changes in behaviour due to interaction with deposited graphite. The same applies to thermal expansion, etc. in Figures 5 and 6, which seems to be tested only for the LDPE support.

Reply

In response to the reviewer's comments, we would like to point out that the deposition of a thin  graphite coating (about 2.5 microns) at ambient pressure and temperature, without the use of solvents or surface treatments, is not expected to modify the bulk properties of the polymeric substrate.

Indeed, from DSC curves of LDPE and LDPE + Graphit platelet film shown in the Figure below, can be clearly seen that the peaks are similar in shape and intensity.

The small differences are probably due to the thermal inertia introduced by the coating. Likewise, it has been verified that the thermal linear strain, ε, as a function of the temperature, in the 20-137 °C range of  pristine LDPE and a GP film on LDPE are almost the same.

In any case, one of the aims of this work is to show how the GP coating is able to functionalize a polymeric film and transform it into an extremely sensitive temperature sensor. The operating principle is based on the properties of the particular substrate chosen. The precise characterization of the thermo-mechanical parameters of LDPE used and the accurate determination of the temperature range in which phase transitions occur are thus essential to understand the variations of the thermoresistive properties recorded by the sensor. A generic characterization of LDPE film taken from the literature would not have been sufficient to investigate the properties of GP/substrate system.

Reviewer 2 Report

In this work, the authors report on the temperature dependence of electrical resistance in graphite films deposited on glass and low density polyethylene. The morphological, structural and electrical properties characterized by SEM images, Raman spectra, XRD diffractograms and current-voltage characteristics. The authors investigated the thermal resistance of graphite deposited on these two different substrates by studying the resistance trend as a function of temperature in the 20-120oC range, it was found that for glass substrates, the resistance of the films decreased monotonically as a function of temperature by 7%, and for LDPE substrate the film resistance firstly increases more than one order of magnitude in the 20-100°C range, then suddenly decreases to a temeperature between 105 °C and 115 oC. The results are solid and the conclusion is well supported. With answering the following questions, I recommend publication of the manuscript in Coatings:

1. I am curious about the repeatability of the presented data. It would be useful to know how many samples were characterized for each substrate.

2. The spray technique is one of the main innovations of this article, and the authors should provide more details or schematic diagrams of the modified technique for better repetition and understanding.

3. The author mentions that the thickness of the sample is 2.5um in line 7 on page 2 of the article, and the characterization of the thickness is determined from the reference [23]. We know that the thickness of graphite has a great influence on its resistance, so it is very necessary for the author to give a direct characterization of the thickness in this article, such as AFM or TEM.

4. It is well known that stress has a great influence on the resistance of graphite films on flexible substrates. Therefore, we suggest that the authors test the Raman spectra of graphite films at different temperatures and analyze the changes of the position in G peak, thereby ruling out that the abnormal temperature-dependent behavior of resistance is caused by stress.

5. From the Raman spectra in figure 1, it can be seen that there are strong defect peak (D peak) on both substrate, which is inconsistent with the high-quality graphite mentioned by the author. Why are the defects so strong and whether the defects are uniform distributed? It is recommended that the author give out Raman mapping.

Author Response

Comments on coatings-1844614 manuscript                                                                         

Response to Reviewer 2

1. General comments of Reviewer 2

In this work, the authors report on the temperature dependence of electrical resistance in graphite films deposited on glass and low density polyethylene. The morphological, structural and electrical properties characterized by SEM images, Raman spectra, XRD diffractograms and current-voltage characteristics. The authors investigated the thermal resistance of graphite deposited on these two different substrates by studying the resistance trend as a function of temperature in the 20-120oC range, it was found that for glass substrates, the resistance of the films decreased monotonically as a function of temperature by 7%, and for LDPE substrate the film resistance firstly increases more than one order of magnitude in the 20-100°C range, then suddenly decreases to a temeperature between 105 °C and 115 oC. The results are solid and the conclusion is well supported. With answering the following questions, I recommend publication of the manuscript in Coatings:

Response to the general comments:

We have really appreciated the kind review and modified the earlier manuscript according to the valuable comments and recommendations of the reviewer.

All the modified paragraphs of the text are highlighted in yellow below and in an additional copy of the revised manuscript uploaded as source file coatings-1844614rev.docx

2. Reviewer’s  comment

I am curious about the repeatability of the presented data. It would be useful to know how many samples were characterized for each substrate.

Reply

We  would like to thank the reviewer for his/her comments that helped to improve the manuscript. In response to the reviewer’s question we specify that four runs of deposition were performed and for each run a pair of graphite films was obtained at room temperature (RT) under the same deposition conditions on glass and LDPE substrates. Therefore, we characterized four GP films deposited on LDPE and four films on glass. The behaviors of the structural, morphological and electrical properties of the representative samples shown in the manuscript were confirmed in all the investigated samples.

Changes to the text are reported below in the revised text and highlighted in yellow.

Pag.2 line 22 

Previous text

Large area thin films, about 15 cm x 15 cm of graphite-based material were deposited on glass and LDPE substrates by spray technology, using a commercial lacquer, Graphit 33 (from Kontakt Chemie, Zelecity, Belgium).

Revised text                                              

Large area thin films of graphite-based material were deposited on glass and LDPE polymer substrates by spray technology, using a commercial lacquer, Graphit 33 (from Kontakt Chemie, Zelecity, Belgium).  LDPE substrates, of size 15 cm x 15 cm x 90 mm, were produced by Sabic (Saudi Arabia, 99.77% by weight, Mn=28 0000 gmol−1, melt flow index of 7 g/10 min and crystalline fraction (Xc) of 32.8%),  while glass substrates (SiO2)  of size 2.5 cm x 7.6 cm x 1 mm, were fabricated by Pearl inc. (West Hollywood, CA, USA).

The figure 1 shows the scheme of the apparatus used for the spray technique.

Figure 1. Scheme of the apparatus used for the spray deposition technique.

The spray nozzle was directed to a distance of 20 cm from plane containing the substrates to produce a uniform deposition by full cone spot, and horizontally moved so that a graphite film could completely cover the surface of the substrates. Four runs of depositions were performed and for each run a pair of graphite films was obtained at room temperature (RT) under the same deposition conditions on glass and LDPE substrates.

Before lacquer spraying no cleaning treatment was performed on LDPE substrates, while  glass substrates were cleaning by acetone.

3. Reviewer’s comment

The spray technique is one of the main innovations of this article, and the authors should provide more details or schematic diagrams of the modified technique for better repetition and understanding.

Reply

We  would like to thank the reviewer for  his/her suggestion. Accordingly, the scheme of the apparatus used for the spray deposition technique is added in the revised manuscript as Figure 1 and the changes to the text are highlighted in yellow.

Pag.2 line 22 

Previous text

Large area thin films, about 15 cm x 15 cm of graphite-based material were deposited on glass and LDPE substrates by spray technology, using a commercial lacquer, Graphit 33 (from Kontakt Chemie, Zelecity, Belgium).

Revised text                                              

Large area thin films of graphite-based material were deposited on glass and LDPE polymer substrates by spray technology, using a commercial lacquer, Graphit 33 (from Kontakt Chemie, Zelecity, Belgium).  LDPE substrates, of size 15 cm x 15 cm x 90 mm, were produced by Sabic (Saudi Arabia, 99.77% by weight, Mn=28 0000 gmol−1, melt flow index of 7 g/10 min and crystalline fraction (Xc) of 32.8%),  while glass substrates (SiO2)  of size 2.5 cm x 7.6 cm x 1 mm, were fabricated by Pearl inc. (West Hollywood, CA, USA).

The figure 1 shows the scheme of the apparatus used for the spray technique.

Figure 1. Scheme of the apparatus used for the spray deposition technique.

4. Reviewer’s  comment

The author mentions that the thickness of the sample is 2.5um in line 7 on page 2 of the article, and the characterization of the thickness is determined from the reference [23]. We know that the thickness of graphite has a great influence on its resistance, so it is very necessary for the author to give a direct characterization of the thickness in this article, such as AFM or TEM.

Reply

We agree with the reviewer’s comment, however  the required information are included in our previous paper  and the editor recommended not to exceed 30% of the repetitions, so we preferred to cite this paper [Ref. 27 of the manuscript] for what concerns the TEM measurements carried out for the determination of the GP film thickness.

 5. Reviewer’s  comment

 It is well known that stress has a great influence on the resistance of graphite films on flexible substrates. Therefore, we suggest that the authors test the Raman spectra of graphite films at different temperatures and analyze the changes of the position in G peak, thereby ruling out that the abnormal temperature-dependent behavior of resistance is caused by stress.

Reply

We are grateful to the reviewer for  suggesting to analyze the Raman spectra of the coating at different temperatures. However, these measurements require a specific accessory to be coupled to the Raman apparatus currently not available in our laboratory. We are planning to investigate this issue in future work.

6. Reviewer’s comment

From the Raman spectra in figure 1, it can be seen that there are strong defect peak (D peak) on both substrate, which is inconsistent with the high-quality graphite mentioned by the author. Why are the defects so strong and whether the defects are uniform distributed? It is recommended that the author give out Raman mapping.

Reply

We  would like to thank the reviewer for the valuable comments that helped us to improve the manuscript. We would point out that the D peak is generated by the edges of the graphite platelets [REF. 31 in the manuscript] and the quality of the graphite network is witnessed by the D and G peak width. Raman spectra analysis indicates that the D and G peaks are sharp and well separated as it occurs in good quality graphite.

Changes to the text are reported below in the revised text and highlighted in yellow. 

As for the Raman mapping of the samples, we are not able to perform them in this period. We are planning to carry out these measurements in a future work.                                                                                                    

Pag. 3 Line 20

Previous text

The Raman spectra of two representative samples prepared on glass and LDPE substrates are plotted in Figures 1a) and 1b), respectively. 

Spectra analysis denotes that the deposited films have a graphite-like structure with similar features on both substrates: a narrow D peak, a sharp and narrow G band, with a shoulder (D*) due to the presence of defects [30, 31]. The peak intensity ratio, Id/Ig, averaged over the sampled points, are 0.55 and 0.80 for the films on glass and LDPE, respectively.  It reveals that the films are obtained from good quality graphite-based material and exhibit slightly higher defect density when deposited on LDPE.

Three other peaks can be detected in both spectra at higher wavenumbers between 2500 and 3500 cm−1: an intense left shouldered 2D peak indicating a multilayered structure, a D + G peak, and a 2G peak typical in width and intensity of graphite material. No substrate signal is detected indicating that the film’s thickness is higher than the penetration length of the laser radiation used in the Raman apparatus.

Revised text

The Raman spectra of two representative samples prepared on glass and LDPE substrates are plotted in Figures 1a) and 1b), respectively. No substrate signals are detected as the films thicknesses are higher than the penetration length of the laser radiation used in the Raman apparatus, a few hundred nanometers. Hence, the Raman spectra features are due to the graphite platelets embedded in the lacquer. The features in the 1000-1800 cm-1 region are quite similar in the two spectra: a narrow D peak, a sharp and narrow G band, with a shoulder (D*) due to the presence of defects [30]. The peaks are sharp, indicating that the graphite platelets are of good crystalline quality. The peak intensity ratio, Id/Ig, averaged over the sampled points, are 0.55 and 0.80 for the films on glass and LDPE, respectively. It is worthwhile to note that the D peak is generated by the platelet edges [31], indeed  in the case of a platelet slightly larger than the beam and perpendicular to it, no edges will be excited and no D peak will be detected [31]. On the other end, for bended platelet not perpendicular to the beam, some edges will  be illuminated producing a D peak. The results indicate that more edges fall into the area swept by the laser beam in the LDPE case. This is consistent with a slightly more disorder in orientation of the platelets on the LDPE substrate. Three other peaks can be detected in both spectra at higher wavenumbers between 2500 and 3500 cm−1: an intense left shouldered 2D peak indicating a multilayered structure, a D + G peak, and a 2G peak typical in width and intensity of graphite material.

Reviewer 3 Report

This article is a study of the thermoelectric properties of a graphite layer deposited on two different substrates - glass and polyethylene. The authors conducted a study of the surface of the graphite layer deposited on different substrates and the temperature dependence of the electrical resistance. Additionally, some properties of the polyethylene substrate have been investigated. Despite the fact that the idea of the study is not particularly new, the article is made at a good research level, using modern technologies and methods. The article is easy to read. There are a number of minor typos that I am sure will be corrected during the editing process. I believe that this article can be published in this special issue after the authors answer some questions and supplement the article with some information:

 1. I think that it is necessary to add the characteristics of the substrates, including the chemical composition of the glass, the dimensions of the substrates, the thickness of the substrates, etc. Does the film completely cover the substrate?

2. How firmly the film is held on the substrate? Was there any additional processing of the substrates before applying the layer? For example, glass etching?

3. Could the authors comment on the prospects of the obtained results for application and further research?

Author Response

Comments on coatings-1844614 manuscript        

Response to Reviewer 3

1. General comments of Reviewer 3

This article is a study of the thermoelectric properties of a graphite layer deposited on two different substrates - glass and polyethylene. The authors conducted a study of the surface of the graphite layer deposited on different substrates and the temperature dependence of the electrical resistance. Additionally, some properties of the polyethylene substrate have been investigated. Despite the fact that the idea of the study is not particularly new, the article is made at a good research level, using modern technologies and methods. The article is easy to read. There are a number of minor typos that I am sure will be corrected during the editing process. I believe that this article can be published in this special issue after the authors answer some questions and supplement the article with some information: 

Response to the general comments:

We have really appreciated the kind review and modified the earlier manuscript according to the valuable comments and recommendations of the reviewer.

All the modified paragraphs of the text are highlighted in yellow below and in an additional copy of the revised manuscript uploaded as source file coatings-1844614rev.docx.

2. Reviewer’s  comment

I think that it is necessary to add the characteristics of the substrates, including the chemical composition of the glass, the dimensions of the substrates, the thickness of the substrates, etc. Does the film completely cover the substrate?.

Reply

We would like to thank the reviewer for his/her comments that helped us to improve the manuscript. Accordingly, we have added more information on the substrate characteristics and deposition conditions in the revised text and highlighted in yellow as reported below.

Pag.2 line 22 

Previous text

Large area thin films, about 15 cm x 15 cm of graphite-based material were deposited on glass and LDPE substrates by spray technology, using a commercial lacquer, Graphit 33 (from Kontakt Chemie, Zelecity, Belgium).

Revised text                                              

Large area thin films of graphite-based material were deposited on glass and LDPE polymer substrates by spray technology, using a commercial lacquer, Graphit 33 (from Kontakt Chemie, Zelecity, Belgium).  LDPE substrates, of size 15 cm x 15 cm x 90 mm, were produced by Sabic (Saudi Arabia, 99.77% by weight, Mn=28 0000 gmol−1, melt flow index of 7 g/10 min and crystalline fraction (Xc) of 32.8%),  while glass substrates (SiO2)  of size 2.5 cm x 7.6 cm x 1 mm, were fabricated by Pearl inc. (West Hollywood, CA, USA).

The figure 1 shows the scheme of the apparatus used for the spray technique.

Figure 1. Scheme of the apparatus used for the spray deposition technique.

The spray nozzle was directed to a distance of 20 cm from plane containing the substrates to produce a uniform deposition by full cone spot, and horizontally moved so that a graphite film could completely cover the surface of the substrates. Four runs of depositions were performed and for each run a pair of graphite films was obtained at room temperature (RT) under the same deposition conditions on glass and LDPE substrates.

Before lacquer spraying no cleaning treatment was performed on LDPE substrates, while  glass substrates were cleaning by acetone.

3. Reviewer’s comment

How firmly the film is held on the substrate? Was there any additional processing of the substrates before applying the layer? For example, glass etching?

Reply

We would like to thank the reviewer for his/her comments that helped us to improve the manuscript. Accordingly, we have added more information on the adhesion mechanism of graphite coating to the different substrates in the revised text and highlighted in yellow as reported below.

We would like to point out that before the spray deposition no cleaning treatment was performed on LDPE substrates, while glass substrates were cleaning by acetone.

After spraying the lacquer on LDPE and glass substrates the following adhesion mechanism takes place. Graphite platelets are able to stick to the surface of non-polar materials like polyethylene because physical interactions are possible between them. Indeed, Graphene sheets interact inside the graphite crystal by p-p physical bonds, that is a special dipole-dipole interaction and the C-H bond in the methylene groups of LDPE also may give dipole-dipole interaction with the graphite. Owing to the pseudo-planar shape and tendential iso-orientation of graphite nanoplatelets, the number of interaction with the flat LDPE film surface is very large and therefore a good LDPE-Graphit 33 adhesion takes place.

Pag.2 line 22 

Previous text

Large area thin films, about 15 cm x 15 cm of graphite-based material were deposited on glass and LDPE substrates by spray technology, using a commercial lacquer, Graphit 33 (from Kontakt Chemie, Zelecity, Belgium).

Revised text                                              

Large area thin films of graphite-based material were deposited on glass and LDPE polymer substrates by spray technology, using a commercial lacquer, Graphit 33 (from Kontakt Chemie, Zelecity, Belgium).  LDPE substrates, of size 15 cm x 15 cm x 90 mm, were produced by Sabic (Saudi Arabia, 99.77% by weight, Mn=28 0000 gmol−1, melt flow index of 7 g/10 min and crystalline fraction (Xc) of 32.8%),  while glass substrates (SiO2)  of size 2.5 cm x 7.6 cm x 1 mm, were fabricated by Pearl inc. (West Hollywood, CA, USA).

The figure 1 shows the scheme of the apparatus used for the spray technique.

Figure 1. Scheme of the apparatus used for the spray deposition technique.

The spray nozzle was directed to a distance of 20 cm from plane containing the substrates to produce a uniform deposition by full cone spot, and horizontally moved so that a graphite film could completely cover the surface of the substrates. Four runs of depositions were performed and for each run a pair of graphite films was obtained at room temperature (RT) under the same deposition conditions on glass and LDPE substrates.

Before lacquer spraying no cleaning treatment was performed on LDPE substrates, while  glass substrates were cleaning by acetone.

Pag. 3 Line 18

Previuos text

Films deposited on different substrates by spraying Graphit33 lacquer under the same preparation conditions are investigated in their morphological, structural and thermoresistive properties. 

Revised text

Films deposited on different substrates by spraying Graphit33 lacquer under the same preparation conditions are investigated in their morphological, structural and thermoresistive properties. After spraying the lacquer, a good adhesion of the graphite film takes place on LDPE substrate because graphite platelets interact with the surface of non-polar materials, like polyethylene. On the other hand the polymeric binder present in the lacquer allows the adhesion of the graphite film  to the glass substrate.                                   

4. Reviewer’s  comment

Could the authors comment on the prospects of the obtained results for application and further research?

Reply

We would like to thank the reviewer for his/her comments that helped us to improve the manuscript. Accordingly, we have modified the text in the revised manuscript and highlighted in yellow as reported below.

In response to the reviewer's comments, we would like to point out that the aim of this work is to measure the electro-thermal properties of a film of graphite platelets deposited on glass and LDPE substrates by mean of a spray technology. These composite structures have potential applications for flexible electronics, including temperature sensors and self-switching components. In particular, the use of polymer substrates is an emerging technology that has great potential for the production of large-area, lightweight, and mechanically flexible electronics at low cost. Their manufacture requires the integration of materials with different mechanical and thermal properties, from rigid conductive coatings to compliant polymer substrates. It is essential that these materials have consistent electrical and mechanical behavior over a wide range of temperature and load conditions. Understanding the mechanical and thermal properties of such composite structures is critical to bringing the technology to mass production.

Pag. 2 Line 18

Previous text

The thermo-resistive properties of GP deposited on LDPE and glass substrates are investigated by the resistance trends as a function of the temperature in the 20-120 °C range. In the case of films on LDPE, variations in the thermal expansion coefficient and phase transitions occurring in the polymer substrates are taken into account in the discussion.

Revised text

The thermo-resistive properties of GP deposited on LDPE and glass substrates are investigated by the resistance trends as a function of the temperature in the 20-120 °C range. In the case of films on LDPE, variations in the thermal expansion coefficient and phase transitions occurring in the polymer substrates are taken into account in the discussion.

Understanding the dependence of thermorestive properties on the thermo-mechanical ones of such composite structures is crucial in view of large-scale production.

Round 2

Reviewer 1 Report

The work has improved enough for publication.